# The Impact of Clinical Pharmacist-Driven Weekend Antimicrobial Stewardship Coverage at a Quaternary Hospital

**DOI:** 10.3390/antibiotics13100974

**Published:** 2024-10-16

**Authors:** Hazem Elrefaei, Wasim S. El Nekidy, Rama Nasef, Manal Motasem, Yara Mkarim, Osama Al Quteimat, Mohamed Hisham, Rami Ismail, Emna Abidi, Claude Afif, Rania El Lababidi

**Affiliations:** 1Department of Pharmacy Services, Cleveland Clinic Abu Dhabi, Abu Dhabi P.O. Box 112412, United Arab Emirates; elnekiw@clevelandclinicabudhabi.ae (W.S.E.N.); nasefr@clevelandclinicabudhabi.ae (R.N.); mmotasem@capital-health.ae (M.M.); or yara.makarem@gmail.com (Y.M.); alquteo@clevelandclinicabudhabi.ae (O.A.Q.); hishamm@clevelandclinicabudhabi.ae (M.H.); ismailr@clevelandclinicabudhabi.ae (R.I.); abidiee@clevelandclinicabudhabi.ae (E.A.); ellabar@clevelandclinicabudhabi.ae (R.E.L.); 2Cleveland Clinic Lerner College of Medicine, Cleveland, OH 44106, USA; 3Infectious Diseases Department, Cleveland Clinic Abu Dhabi, Abu Dhabi P.O. Box 112412, United Arab Emirates; afifc@clevelandclinicabudhabi.ae

**Keywords:** antimicrobial stewardship, pharmacist driven, weekends, clinical pharmacy, pharmacist interventions, antibiotics

## Abstract

Background and objective: Extending a consistent pharmacy antimicrobial stewardship weekend service was a newly implemented initiative. We sought to evaluate the impact of incorporating an Infectious Diseases (ID)-trained clinical pharmacist into an antimicrobial stewardship program (AMS) during weekends. Results: The number of documented interventions was 451 on 362 patients compared to 115 interventions on 108 patients during the pre-implementation period (*p* = 0.04), with interventions primarily targeting Watch antibiotics, as classified by the WHO AWaRe classification. A reduction in the LOS was observed, with a median of 16 days (8–34) during the post-implementation period compared to 27.5 days (10–56) during the pre-implementation period (*p* = 0.001). The median DOT increased during the post-implementation period to 8 (6–11), versus the increase to 7 (4–11) during the pre-implementation period (*p* ≤ 0.001). Finally, there was no significant difference observed in healthcare-associated CDI and infection-related readmission. Methods: This is a retrospective single-center, pre–post quasi-experimental study. Data including the documented pharmacist interventions were collected from the electronic medical record (EMR), the pre-implementation phase was in 2020, and post-implementation was in 2021. The primary outcome was to identify the number of AMS interventions through prospective audit and feedback review analysis. Secondary outcomes included antibiotic days of therapy (DOT), length of hospital stay (LOS), healthcare-associated *Clostridioides difficile* infection (CDI), and infection-related readmission. Conclusions: The pharmacist-driven weekend AMS is an opportunity for pharmacists to intervene and optimize patients’ care plans. This initiative demonstrated significant increased AMS-related interventions, promoted judicious antimicrobial use, and contributed to a reduced length of hospital stay. Our findings need to be replicated in a larger prospective study.

## 1. Introduction

Antimicrobial resistance (AMR) is a growing concern worldwide and is now considered one of the most significant threats to public health. The overuse and misuse of antibiotics have been linked to increased rates of adverse drug events and the development of multidrug-resistant microorganisms, which can lead to prolonged hospital stays, greater morbidity and mortality, and higher healthcare costs [1,2]. Studies have reported that as much as 40–50% of antibiotic prescriptions in hospitals, worldwide, might be unnecessary. Even when antibiotics are necessary, they are often prescribed with excessive broad-spectrum coverage or for longer durations than needed [3,4].

Around 20% of hospitalized patients who receive antibiotics experience adverse events, and antibiotic use can impact not only the individual patient’s microbiome but contribute to antibiotic resistance in institutions and communities. To address the growing threat of antimicrobial resistance, the World Health Organization (WHO) has developed a practical resource to guide healthcare professionals to implement and strengthen antimicrobial stewardship programs (AMSs) across various healthcare settings [5]. As part of their ongoing efforts, the WHO introduced the AWaRe classification system, which guides prescribers to prioritize “Access” antibiotics and employ responsible prescribing practices for “Watch” and “Reserve” options [6]. In line with global efforts, the Infectious Diseases Society of America (IDSA) published a guideline for implementing an antibiotic stewardship program [7]. The aim of this guidance is to optimize antibiotic prescribing and reduce harm caused by unnecessary antibiotic use. The IDSA emphasizes that successful hospital antimicrobial stewardship programs require a collaborative approach and recommends appointing a pharmacist as a co-leader of the stewardship program [7]. This reflects the valuable contribution that pharmacists can make to optimizing antibiotic use.

Several studies have reported that pharmacists can play a key role in assuring the optimal use of antimicrobial agents, monitoring and auditing the prescriptions, and educating health professionals [8].

Pharmacist-led antimicrobial stewardship programs (AMSs) have proven to enhance clinical outcomes, reduce untoward outcomes of antibiotics, and lower healthcare expenses by initiating early antibiotic de-escalation, switching from intravenous to oral therapy, limiting the use of broad-spectrum agents, and recommending appropriate infectious disease (ID) physician consultation [8,9]. However, outside regular business hours, there is often insufficient staffing support with trained personnel. Core stewardship strategies, such as providing prospective audit and feedback during weekends, holidays, and evenings, are limited [10,11]. Furthermore, data on expanding AMS personnel coverage beyond peak hours is limited [12].

Our institution is a 364-bed hospital, which is an extension of a US-based model of care in the Arabian Gulf region. Operational since March 2015, our AMS holds the distinction of being designated a Centre of Excellence by the Infectious Diseases Society of America (IDSA). The program is co-directed by a postgraduate year 2 (PGY2) trained infectious diseases pharmacist, who oversees both the day-to-day clinical activities and the AMS subcommittee, a monthly forum that addresses antimicrobial formulary changes as well as updates to antimicrobial policies, protocols, and guidelines [13]. There are several intermediate- to advanced-level initiatives implemented through the AMS, including prospective audits with intervention and feedback, prior authorization of broad-spectrum antimicrobials, facility-specific antimicrobial guidelines, real-time rapid diagnostics, real-time computerized surveillance systems, and real-time dashboards for AMS metrics [14,15].

During the pre-implementation phase, the AMS was only supported during the weekends with an ID physician and an on-call clinical pharmacist available for consultation, which led to potential inconsistencies in stewardship practices. Our institution implemented the weekend antimicrobial stewardship clinical pharmacist to further prospectively optimize antimicrobial prescribing during the weekend. Hence, the aim of this study is to evaluate the impact/outcomes of the integration of an ID clinical pharmacist into weekends.

## 2. Study Definitions

Broad-spectrum antibiotics are classes of antibiotics that act against an extensive range of disease-causing bacteria by targeting both Gram-positive and Gram-negative bacterial groups. Hospital-acquired infections, also known as healthcare-associated infections (HAI), are nosocomially acquired infections that are typically not present or might be incubating at the time of admission, usually acquired after hospitalization and manifest 48 h after admission to the hospital. Community-acquired infection is any infection acquired in the community; an infection would be classified as community-acquired if the patient had not recently been in a healthcare facility or been in contact with someone who had recently been in a healthcare facility. Antibiotic days of therapy (DOT) is defined as the number of days that a patient receives an antimicrobial agent (regardless of dose); any dose of an antibiotic that is received during a 24 h period represents 1 DOT, and the DOT for a given patient on multiple antibiotics is the sum of the DOT for each antibiotic that the patient has received. Infection-related readmission is defined as hospital readmission occurring because of infection within 30 days. Healthcare-associated CDI is defined as an infection caused by the bacterium *Clostridioides difficile*, which typically occurs in healthcare settings such as hospitals or long-term care facilities; this infection often arises after the normal gut flora has been disrupted, usually due to antibiotic therapy. The infection-related readmission rate (IRRR) is defined as the percentage of patients who were readmitted to the hospital within 30 days of discharge with a new or worsening infection. IRRR is a measure of the quality and safety of care for patients with infections, and it is a key indicator of antimicrobial AMS effectiveness.

## 3. Statistical Analysis

The data normality and distribution were assessed using the Shapiro–Wilk test and by visually inspecting the variable’s distribution (histogram). Data are expressed as the median with the interquartile range (IQR). Proportions were used as descriptive statistics for categorical variables. The Mann-Whiney U test performed a comparison of values between independent continuous variables, while discrete data comparisons were performed by Chi-Square or Fisher’s exact test when the numbers were small. A two-sided *p*-value < 0.05 was used as the criteria to determine statistical significance in this study. The following variables were assumed: a 2-sided significance level of 0.05. All statistical analyses were performed using SPSS statistical package version 29 for Microsoft Windows (IBM, Armonk, NY, USA).

## 4. Results

The pre-implementation phase was composed of 108 subjects, while the post-implementation phase was composed of 362 subjects. Demographics and baseline characteristics were not significantly different between both groups (Table 1).

During the study period, most of the AMS pharmacist interventions were accepted by physicians, with no significant difference observed between the pre-implementation (94%) and post-implementation groups (90%) (*p* = 0.242).

Most interventions during the post-implementation period were de-escalations of antimicrobial therapy (30%), followed by discontinuations of an antibiotic agent due to unnecessary coverage (e.g., vancomycin with no methicillin-resistant *Staphylococcus aureus* (MRSA)) (27.8%), and interventions on inappropriate or unspecified durations of antimicrobial therapy constituted 19.5% of the documented interventions. The most reported interventions during the pre-implementation period were antimicrobial dose optimizations, mainly renal dosage adjustments upon order verification (64%), followed by de-escalation and the initiation of antimicrobial therapy (Table 2). A detailed breakdown of the antibiotics involved in de-escalation (Table 2A) and discontinuation (Table 2B) interventions, categorized by their WHO AWaRe classification, is provided below.

The AMS weekend pharmacist had a higher percentage of interventions on patients with UTIs (30% vs. 24%), IAIs (12% vs. 6.5%), and skin and soft tissue infections (10% vs. 4.5%). RTIs were comparable between the pre- and post-implementation groups (28% vs. 28%), while the pre-implementation group had a higher percentage of patients with bacteremia (36% vs. 20%). Additionally, the post-implementation group had a significantly higher proportion of community-acquired infections (73% vs. 32%) and a higher percentage of patients admitted to acute care units (84% vs. 71%), while the pre-implementation group had more hospital-acquired infections (68% vs. 27%) and a higher percentage of patients in intensive care units (29% vs. 16%) (Table 1).

Overall, a significant increase in the number of documented interventions was observed, with 451 interventions documented on 362 patients during the post-implementation period compared to 115 interventions documented on 108 patients during the pre-implementation period (*p* = 0.04). Specifically, the percentage of antimicrobial regimen de-escalation interventions increased significantly from 18% pre-AMS to 30% post-AMS (*p* < 0.001). The percentage of discontinuation interventions increased significantly from 1.75% pre-AMS to 27.8% post-AMS (*p* < 0.001). The percentage of duration-of-therapy interventions increased significantly from 1.75% pre-AMS to 19.5% post-AMS (*p* < 0.001). The percentage of bug–drug mismatch interventions increased significantly from 1.75% pre-AMS to 8.2% post-AMS (*p* = 0.006).

The percentage of dose optimization interventions decreased significantly from 64% pre-AMS to 7.1% post-AMS (*p* < 0.001). The percentage of initiate therapy interventions decreased from 11% pre-AMS to 4.7% post-AMS (*p* = 0.058). The percentage of IV to PO interventions increased from 0% pre-AMS to 2% post-AMS (*p* = 0.127). The percentage of lab request interventions remained relatively unchanged between pre-AMS (1.75%) and post-AMS (0.7%) (*p* = 0.324) values (Table 2).

During the post-implementation prospective audit and feedback review on week-ends, the most common antimicrobial agents involved in interventions were piperacillin–tazobactam (44%), vancomycin (15%), meropenem (11%), ertapenem (8%), and cefepime (8%) (Figure 1). Notably, most of these interventions targeted antibiotics classified as ‘Watch’ by the WHO AWaRe classification, with a particular focus on Piperacillin-Tazobactam, Meropenem, and Vancomycin (Table 2A,B). 

A significant reduction in the LOS was observed, with a median (IQR) of 16 days (8–34) during the post-implementation period compared to 27.5 days (10–56) during the pre-implementation period (*p* = 0.001). On the contrary, the median (IQR) of the total DOT increased during the post-implementation period to 8 (6–11), versus to 7 (4–11) during the pre-implementation period (*p* < 0.001). No statistically significant differences were observed in healthcare-associated CDI, the percentage of patients discharged on antibiotics, home regimen DOT, and infection-related readmission (Table 3).

Implementing weekend AMS pharmacist coverage led to a substantial reduction in overall costs. During the pre-implementation period, cost savings primarily resulted from antimicrobial de-escalation interventions, averaging $277 per patient. Post-implementation, however, saw a shift in savings mechanisms. De-escalation from broad-spectrum to narrow-spectrum antimicrobials yielded an average saving of $157 per patient, while the discontinuation of unnecessary vancomycin contributed a significant $202 saving per patient. Overall, the average cost savings observed during the post-implementation period reached $28,493, a considerable increase compared to the $4432 saved during the pre-implementation period.

We estimated direct drug cost savings by comparing the average daily cost of antibiotic therapy per patient before and after implementation, using our hospital’s formulary pricing for 2020 (pre-implementation) and 2021 (post-implementation). This approach isolates cost differences attributable to pharmacist interventions, such as the de-escalation and discontinuation of unnecessary therapy. Cost saving calculations have multiple limitations as saving associated with interventions that impact a patient’s outcome, reduce antibiotics-related adverse events, and shorten the hospital LOS are challenging to estimate, as opposed to costs associated with drug acquisition.

## 5. Discussion

In the present study, we were able to describe our experience in expanding the AMS services to ensure the continuity of patient-centric care throughout the whole week. We observed a significant increase in the AMS-related interventions documented throughout the year, even though the service was provided for 4 h per day on weekends. The antibiotic regimen DOT was numerically but not significantly lower after service model implementation. Additionally, the service model implementation was associated with a significantly shorter hospital LOS. Finally, the service model was not associated with a significant difference in CDI percentages or infection-related readmission rate. Unlike Bohn, Siegfried, Lacy, and colleagues’ studies that relied on PGY-2 ID pharmacy residents to cover weekends, our study relied on an ID board-certified clinical pharmacists to perform the AMS prospective audit and feedback [10,11,16].

Several studies have shown that expanding antimicrobial stewardship (AMS) services to include weekends can significantly increase the number of interventions performed. Bohn et al. conducted a quasi-experimental study in a large US academic medical center. They investigated the impact of weekend service expansion on AMS activities. The study included 72 subjects before implementation and 59 subjects after implementation over a 13-week period. The authors found that the additional weekend service resulted in an extra 1258 AMS activities [10]. Siegfried et al. conducted another study in a large urban hospital. This study assigned PGY2 pharmacy residents specializing in infectious disease (ID) or critical care to provide weekend AMS coverage over a 12-month period. These residents documented a total of 1443 interventions during their weekend coverage. Notably, 1000 (69%) of these interventions stemmed from a 72 h prospective audit and feedback review process [11]. Our findings align with these studies, demonstrating a significant increase in AMS interventions from 115 per year to 451 per year (*p* = 0.04). However, our study duration was significantly longer over the span of 2 years compared to 4 months total for Bohn et al. Our study highlights the positive impact of pharmacist interventions on AMS outcomes. Notably, physician acceptance of these interventions remained consistently high both before and after AMS implementation (94% vs. 90%, *p* = 0.242). The service led to a significant shift in intervention types. Post-implementation, we observed a significant increase in interventions focused on de-escalation (30% vs. 18%, *p* < 0.001), discontinuation due to unnecessary coverage (27.8% vs. 1.75%, *p* < 0.001), antimicrobial duration-of-therapy optimization (19.5% vs. 1.75%, *p* < 0.001), and interventions addressing bug–drug mismatches (8.2% vs. 1.75%, *p* = 0.006). These interventions often targeted WHO AWaRe “Watch” antibiotics such as piperacillin-tazobactam, meropenem, cefepime, and vancomycin [6]. This shift towards optimizing the use of critically important antimicrobials aligns with core AMS principles and is crucial for combating the global threat of antimicrobial resistance. By reducing unnecessary use, we aim to preserve the effectiveness of these antibiotics for the future. While dose optimization interventions decreased (64% to 7.1%, *p* < 0.001) post-implementation, this is attributed to most initial dose adjustments being performed by clinical pharmacists during order verification, a process that precedes AMS pharmacist review; the AMS pharmacist may further adjust antimicrobial doses later based on changes in patients’ condition, though these adjustments are less frequent than the initial optimizations performed at order verification. These findings collectively demonstrate a shift towards a more comprehensive AMS approach, optimizing not just the dose, but also the choice, duration, and appropriateness of antimicrobial therapy.

Bohn and colleagues, in a retrospective study on 131 patients over 13 weekends, assessed the percentage of overall DOT attributed to weekends before and after implementation; they did not observe a statistical difference in overall DOT upon evaluating weekend pharmacy AMS services 20,551/82,982 (24.1%) pre-implementation vs. 20,604/85,165 (24.8%) post-implementation [10]. Siegfried et al. also assessed the impact of PGY2 resident AMS weekend coverage on overall antibiotic use (antimicrobial utilization) over a year. Their results showed a decrease in total antibiotic use, from 799.3 DOT/1000 patient days before implementation to 740.7 DOT/1000 patient days after implementation (*p* = 0.08). However, this decrease was not statistically significant. One possible explanation for the non-significant result is that Siegfried et al. measured total antibiotic use across all types of infections and medications. This broad approach might have masked the impact of AMS interventions on specific antibiotics or particular infectious syndromes [11]. By contrast, we observed that the median (IQR) DOT, statistically significantly, increased by 1 day during the post-implementation period to 8 (6–11), versus 7 (4–11) during the pre-implementation period (*p* ≤ 0.001). This could be attributed to multiple factors: of these, patients with more complex infections or infections caused by resistant organisms might need more aggressive treatment with a combination of antimicrobial therapy or longer courses of treatment to ensure that the infection is completely eradicated. It is important to highlight that the goal of an AMS is not to reduce the overall number of antibiotic DOT. Rather, it is to ensure that antibiotics are used rationally, aiming to help clinicians to make informed decisions about the best course of therapy for their patients.

Bohn and colleagues found no significant difference in the length of stay (LOS) between the pre-implementation group (median of 13 days, IQR 6–22) and the post-implementation group (median of 14 days, IQR 6–29) (*p* = 0.396) [10]. This lack of difference might be attributed to the relatively short intervention duration of 13 weekends. In contrast, Wang and colleagues conducted a retrospective quasi-experimental study in two independent hepatobiliary surgery and respiratory wards in a tertiary general hospital in China. Over a 10-month intervention phase, they observed that pharmacist-led antimicrobial stewardship (AMS) contributed to a reduction in the average LOS in the hepatobiliary ward by 3.234 days (*p* = 0.006) [17]. Similarly, our study observed a significant reduction in LOS, with a median reduction of 11.5 days during the 12-month post-implementation phase (*p* = 0.001). To our knowledge, this is the first study to demonstrate an association between pharmacist-led AMS interventions on weekends and a reduction in LOS.

The observed reduction in length of stay (LOS) is likely influenced by a combination of factors. While the AMS intervention likely played a role in optimizing antibiotic use and accelerating infection resolution, it is essential to acknowledge that potential confounding variables might have influenced the results. Notably, the pre- and post-implementation cohorts exhibited significant differences in patient characteristics (Table 1). This includes variations in infection site (*p* = 0.007), infection type (*p* < 0.001), and patient location (*p* = 0.004), which could independently influence LOS. For example, the pre-implementation group had a higher proportion of patients with bacteremia, often requiring longer treatment. Conversely, the post-implementation group had a higher prevalence of infections associated with shorter hospital stays, such as urinary tract, intra-abdominal, and skin and soft tissue infections. Furthermore, the shift towards community-acquired infections and increased admissions to acute care units in the post-implementation period may also have influenced LOS. While a significant reduction in LOS was observed, attributing the entire effect solely to the AMS intervention would be an overgeneralization. It is important to note that this reduction likely reflects a combination of factors, including optimized antibiotic use, improved discharge planning, and other multidisciplinary efforts.

It is crucial to acknowledge that our pharmacist-led AMS initiative was one of several factors contributing to the observed outcomes. A multidisciplinary approach was employed during the study period to reduce length of stay and optimize antimicrobial use. This included initiatives such as the development of indication-specific order sets to guide physicians in prescribing the most appropriate empiric antimicrobial therapy, the implementation of 72 h antibiotic timeouts to prompt reassessment of antibiotic necessity and facilitate timely discontinuation, automated infectious disease consults for restricted antimicrobial orders to ensure expert review for high-risk cases, and outpatient parenteral antibiotic therapy (OPAT) referrals to allow eligible patients to transition to outpatient treatment. Additionally, discharge planning was refined with home nursing services to support patients requiring ongoing intravenous antibiotics at home, and stable patients were transferred to long-term care facilities, freeing up acute care beds and facilitating appropriate care transitions. Finally, daily multidisciplinary team meetings were implemented for discharge planning, fostering the timely identification and resolution of discharge barriers. While a significant reduction in LOS was observed, this can be partially attributed to the multidisciplinary efforts implemented alongside the pharmacist-led weekend AMS service. These efforts synergistically contributed to optimizing antimicrobial use, improving patient outcomes, and facilitating more timely discharges.

Surprisingly, there was no statistically significant difference in the percentage of healthcare-associated CDI, or the percentage of patients discharged on antibiotics or IRRR between the two periods. Several factors might have contributed to this finding. The sample size, while larger in the post-implementation group, may have been insufficient to detect subtle differences in readmission rates. Additionally, the shift in the patient population towards more community-acquired infections and acute care admissions in the post-implementation group could also have independently influenced readmission rates. Finally, other complex factors beyond antibiotic use, like patient comorbidities and post-discharge care, likely play a significant role in readmissions, potentially overshadowing the impact of the AMS interventions alone. However, AMS pharmacists may contribute to IRRR by promoting the appropriate use of antimicrobials, educating patients on medication adherence importance and follow-up care coordination.

A systematic review of antimicrobial stewardship programs (AMSs) involving clinical pharmacists in small and medium-sized hospitals shows a significant decrease in the consumption and cost of antimicrobials [18]. Multiple studies have reported a notable increase in ID pharmacist participation in AMSs and an impact on optimizing antimicrobial therapy after their interventions [18]. While a comprehensive cost–benefit analysis is beyond the scope of this retrospective study, and the literature on cost–benefit analysis methods specifically applied to AMS interventions in the UAE is limited, our findings suggest a positive economic impact associated with the pharmacist-led weekend AMS. Our analysis revealed a considerable increase in cost savings from $4432 in the pre-implementation period to $28,493 post-implementation, representing a 542.9% increase. This difference can be attributed to a shift in the types of interventions driving cost savings. Pre-implementation savings primarily stemmed from de-escalation, averaging $277 per patient. Post-implementation, de-escalation to narrower-spectrum antibiotics yielded $157 per patient, while the discontinuation of unnecessary vancomycin emerged as a major driver, saving $202 per patient. It is important to acknowledge that these figures represent a conservative estimate, primarily reflecting direct cost reductions associated with drug acquisition. Quantifying the economic benefits of improved patient outcomes, such as reduced adverse events and shorter lengths of stay, poses a significant challenge. Further research exploring the broader economic landscape, including reduced complications, shorter hospital stays, and improved healthcare system efficiency, has the potential to demonstrate the true magnitude of cost savings associated with pharmacist-driven AMSs.

This study has several strengths, including its focus on addressing the critical gap in weekend antimicrobial stewardship (AMS) services. It provides a detailed description of the intervention, allowing for potential replication in other settings. The study included a comparison between the interventions of clinical pharmacists during the same period in consecutive years, providing a realistic comparison. The study also compares its findings with the existing literature, contextualizing the results and highlighting their significance. By emphasizing patient-centered care and demonstrating cost-effectiveness, this study contributes valuable evidence to support the implementation of pharmacist-driven weekend AMS services.

This study has several limitations, including its retrospective and quasi-experimental design, which increase the risk of bias and confounding variables. The study’s limited sample size hinders the study’s power to detect significant differences in certain outcomes. The inclusion and exclusion criteria may introduce selection bias and limit finding generalizability, as we only included adults admitted to ACUs and ICUs with a defined set of infections. By including only patients on broad-spectrum antimicrobials, the impact of the AMS on patients receiving narrow-spectrum antibiotics was not assessed. Pregnant women were excluded due to the potential risks of antimicrobial therapy on fetal development. Patients with confirmed COVID-19 infection were excluded due to rapidly evolving management strategies during the study period. The study did not include the total number of patients reviewed by the pharmacist per day nor did it measure the impact on overall antimicrobial utilization; we did not have as many interventions since the time of coverage was limited, as compared to other papers published on this topic. The study did not assess the long-term impact of AMS interventions on patient outcomes such as mortality, reinfection rate, and resistance rate. Additionally, the single-center setting and reliance on documented interventions may limit the generalizability of the findings. The study also lacks data on overall antimicrobial utilization and relies on estimated cost savings, which may not capture the full impact of the intervention. Finally, the short intervention duration and concurrent implementation of other initiatives make it difficult to isolate the specific effects of the weekend AMS service.

More research is needed to determine the optimal frequency and duration of AMS interventions.

## 6. Methods

### 6.1. Study Design

This is a single-center, pre–post quasi experimental study. The study was approved by the institution’s Research Ethics Committee. Data were collected retrospectively from the electronic medical record (EMR), a computerized prescriber–order–entry (CPOE) system (Epic^®^ Systems Corporation, Verona, WI, USA). The study period was between January 2020 and December 2021, with the implementation phase taking place in January 2021. Patient demographics, comorbidities, infection type, and patient location acute care units (ACU) vs. intensive care units (ICU) were extracted for baseline characteristic analysis and to aid with the primary end point analysis. The Charlson Comorbidity Index (CCI) score was calculated to estimate the risk of death from comorbid disease for both groups [19]. Data regarding the antibiotic treatment regimen including the duration of therapy, physician response to pharmacist’s recommendations, pharmacist’s interventions, patients’ length of hospital stay, healthcare-associated CDI occurrence, and infection-related readmission were collected for analysis. Data collection for the period before the service implementation took place between 1 January 2020 and 31 December 2020, while the post-implementation phase took place between 1 January 2021 and 31 December 2021.

### 6.2. Outcome Measures

The primary outcome was to evaluate the number and types of antimicrobial-related interventions performed by the AMS pharmacist during weekends following prospective audit and feedback review. Secondary outcomes included antibiotic days of therapy (DOT), length of hospital stay (LOS), healthcare-associated *Clostridioides difficile* infection (CDI), and infection-related readmission rate.

### 6.3. Service Implementation

Prior to 2021, the baseline service was limited in scope. AMS activities over the weekend were part of the clinical pharmacist assignments covering the central inpatient pharmacy. Pharmacist’s AMS interventions on weekends were limited to few activities, primarily regarding medication order review and the verification of new or modified antimicrobial orders, answering physicians’ inquiries through phone calls and upon reviewing antimicrobial dosing-related EMR pharmacist consul orders. Prospective audit and feedback were only performed over the weekdays by the ID clinical pharmacy specialist. An opportunity to provide our patients with better pharmaceutical care was identified, because focusing on order verification means that interventions happen only after the antimicrobial is ordered, potentially delaying optimal therapy adjustments. Moreover, patients who could benefit from antimicrobial de-escalation or discontinuation could not be identified without proactive chart review.

To expand the AMS service to include the weekend, the staffing model was restructured; this was FTE-neutral and incurred no additional costs to the department of the pharmacy. A team of three infectious diseases board-certified clinical pharmacists was formed. These pharmacists received additional in-hospital training through the hospital’s established certificate-based program. The ID-trained clinical pharmacist had a dedicated block of 4 h daily on weekends, from 11 a.m. to 3 p.m. Their responsibilities during this time included prospective audit and feedback for all patients started on broad-spectrum antibiotics, reviewing new blood culture rapid diagnostic test (RDT) results, and reviewing electronic alerts generated by the real-time surveillance software (RLDatix^®^, Chicago, IL, USA) for all other cultures, which improved the time efficiency of this model. Interventions were communicated to the primary team for approval and documented using the EMR iVent system (Epic^®^ Systems Corporation, Verona, WI, USA) under AMS intervention categories. The AMS pharmacist provided a sign-out of open interventions to be followed up by the infectious disease specialist, which resulted in a reduction of workload on Mondays and promoted the continuity of care.

Throughout 2021, we were able to provide the service for 52 weekends; however, the service was interrupted for 6 weeks due to staffing challenges.

### 6.4. Inclusion and Exclusion Criteria

The study population included adults admitted to acute care units (ACUs) and/or intensive care units (ICUs), actively on broad-spectrum antibiotics with the following infections: bacteremia; lower respiratory tract infections, including community-acquired pneumonia (CAP) and/or hospital-acquired pneumonia (HAP); urinary tract infections (UTIs) (pyelonephritis, cystitis); intra-abdominal infections (cholangitis, cholecystitis, appendicitis, diverticulitis); and skin and soft tissue infections (SSTIs) (cellulitis, erysipelas, diabetic foot infections). Pregnant women, patients aged under 18 years, and patients presenting with COVID-19 infection were excluded.

## 7. Conclusions

Overall, our findings demonstrate that AMS pharmacists play a significant role in improving antimicrobial use, patient outcomes, and in improving patient safety. Pharmacists’ involvement in weekend AMSs is an opportunity to bridge the gap in staffing on weekends and ensure that patients receive appropriate antimicrobial therapy, therefore reducing the risk of adverse events and improving patient outcomes. Therefore, healthcare facilities should prioritize the involvement of pharmacists in weekend AMSs to ensure consistent, high-quality care throughout the week.

## Figures and Tables

**Figure 1 antibiotics-13-00974-f001:**
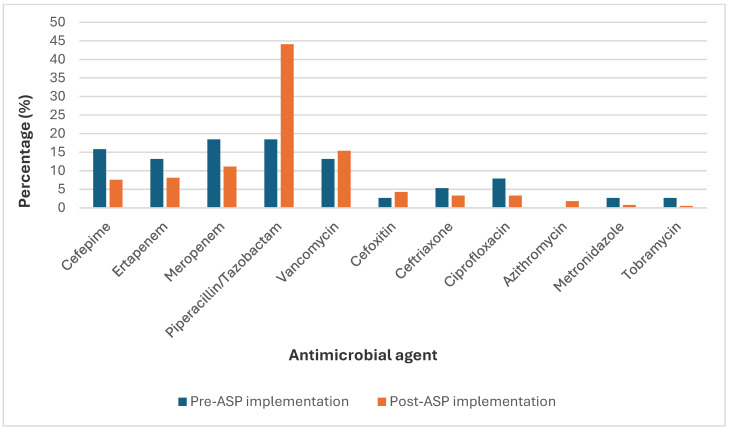
Reported antibiotics in the pharmacist interventions between the pre- and the post-implementation periods.

**Table 1 antibiotics-13-00974-t001:** Patient demographics and baseline characteristics.

Characteristics	Pre-AMS Implementation	Post-AMS Implementation	*p*-Value
	Patients n = 108	Patients n = 362	
Age (years), Median (IQR)	66.5 (48–74)	67 (47–76)	0.711
Males (%)	67 (62)	206 (57)	0.343
Comorbidities (%)			0.091
Multiple	73 (68)	260 (72)	
Diabetes only	5 (4.5)	15 (4)	
Hypertension only	6 (5.5)	5 (1)	
No comorbidities	24 (22)	82 (23)	
Charlson Comorbidity Index Median (IQR)	6 (3–7)	5 (2–7)	0.07
Infection Site (%)			0.007
Bacteremia	39 (36)	74 (20)	
Respiratory Tract Infections	31 (29)	101 (28)	
Urinary Tract Infections	26 (24)	110 (30)	
Intra-abdominal Infections	7 (6.5)	42 (12)	
Skin and Soft Tissue Infections	5 (4.5)	35 (10)	
Infection type (%)			<0.001
Community-acquired	35 (32)	266 (73)	
Hospital-acquired	73 (68)	96 (27)	
Patient Location (%)			0.004
ACU	77 (71)	303 (84)	
ICU	31 (29)	59 (16)	

AMS = Antimicrobial Stewardship Program; ACU = Acute Care Unit; ICU = Intensive Care Unit.

**Table 2 antibiotics-13-00974-t002:** Types of AMS pharmacist interventions on the pre- and post-intervention group.

Intervention	Pre-AMS (n = 115)	Post-AMS (n = 451)	*p*-Value
De-escalation (%)	20 (18)	136 (30)	<0.001
Discontinuation (%)	2 (1.75)	125 (27.8)	<0.001
Duration of Therapy (%)	2 (1.75)	88 (19.5)	<0.001
Bug–Drug Mismatch (%)	2 (1.75)	37 (8.2)	0.006
Dose Optimization (%)	73 (64)	32 (7.1)	<0.001
Initiate Therapy (%)	12 (11)	21 (4.7)	0.058
IV to PO (%)	0 (0)	9 (2)	0.127
Lab Request (%)	2 (1.75)	3 (0.7)	0.324
(**A**) De-escalation interventions by antibiotic and AWaRe classification.			
**Antibiotic**	**AWaRe Classification**	**Pre-AMS (n = 20)**	**Post-AMS (n = 136)**
Piperacillin-Tazobactam (%)	Watch	5 (25)	69 (50.7)
Meropenem (%)	Watch	6 (30)	29 (21.3)
Ertapenem (%)	Access	5 (25)	17 (12.5)
Cefepime (%)	Watch	2 (10)	8 (5.9)
Ceftriaxone (%)	Access	0 (0)	6 (4.4)
Vancomycin (%)	Watch	3 (15)	3 (2.2)
Others (%)	NA	0 (0)	4 (3)
(**B**) Discontinuation interventions by antibiotic and AWaRe classification.			
**Antibiotic**	**AWaRe Classification**	**Pre-AMS (n = 2)**	**Post-AMS (n = 125)**
Vancomycin (%)	Watch	0 (0)	39 (31.2)
Piperacillin-Tazobactam (%)	Watch	1 (50)	37 (29.6)
Meropenem (%)	Watch	0 (0)	8 (6.4)
Ertapenem (%)	Access	0 (0)	7 (5.6)
Cefepime (%)	Watch	0 (0)	8 (6.4)
Amoxicillin/clavulanic acid (%)	Access	0 (0)	5 (4)
Cefoxitin (%)	Access	0 (0)	4 (3.2)
Azithromycin (%)	Watch	0 (0)	4 (3.2)
Ciprofloxacin (%)	Watch	1 (50)	3 (2.4)
Others (%)	NA	0 (0)	10 (8)

NA: Contains more than one classification ex. Watch and access.

**Table 3 antibiotics-13-00974-t003:** Outcomes of pharmacist-driven weekend antimicrobial stewardship on the pre-and post-intervention group.

Variables	Pre-AMS (Patients n = 108)	Post-AMS (Patients n = 362)	*p*-Value
Physician Response Accepted (%)	101 (94)	325 (90)	0.242
Total DOT (Days) Median (IQR)	7 (4–11)	8 (6–11)	0.001
Length of Stay (Days) Median (IQR)	27.5 (10–56)	16 (8–34)	0.001
Discharged on Antibiotics			0.267
No	85 (79)	259 (72)	
Oral	21 (19)	88 (24)	
IV	2 (2)	15 (4)	
Home Regimen DOT (Days) Median (IQR)	7 (5–11)	5 (4–10)	0.211
Healthcare-associated CDI (%)	3 (3)	10 (3)	0.606
Infection-related Readmission (%)	10 (9)	21 (6)	0.204

## Data Availability

The data that support the findings of this study are available upon reasonable request.

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
