# Peer review of "The Impact of Clinical Pharmacist-Driven Weekend Antimicrobial Stewardship Coverage at a Quaternary Hospital"

_antibiotics, 2024, doi:10.3390/antibiotics13100974_

Round 1

Reviewer 1 Report

Comments and Suggestions for Authors

Excellent job.  Some suggestions in the PDF attached.  I would highly suggest to add WHO AMS goals in the introduction as well.

Methods needs to be moved before results.

Comments on the Quality of English Language

Good, a few British spellings vs US spelling

Author Response

Comment 1: Minor editing of English language required.

Response 1: Thank you for your guidance, all were adjusted.

Comment 2: Introduction should include WHO initiative for AMS cited.

Response 2: Thank you, added WHO initiative to introduction, Page 1, line 43.

Kindly find the edited manuscript attached, all changes requested by the three reviewers are highlighted in yellow

the World Health Organization (WHO) has developed a practical resource to guide healthcare professionals to implement and strengthen antimicrobial stewardship programs (AMSs) across various healthcare settings.5

Comment 3: This needs to be moved to after introduction before stats.

Response 3: Thank you, I agree, but this was the journal format requirement.

Kindly find the edited manuscript attached, all changes requested by the three reviewers are highlighted in yellow

Reviewer 2 Report

Comments and Suggestions for Authors

Thanks for giving me an opportunity to review. Always interesting to see research from non-North America/ European countries.

Here are some suggestions:

- Not too sure why the pharmacist's CV is provided.

- The fact that the post-intervention era the median Charlson Comorbidity Index was 5 and the proportion in ICU decreased compared to pre, can we say that the post-intervention cohort was less sick than pre-cohort.

- Is this the reason why we see improvement in certain parameters in Table 3.

- Also there was no statistical difference in those discharged on antibiotics and readmission in the post cohort. How do you explain that?

- Could the section on Cost Benefit Analysis be expanded?

- I find it difficult to prove that all the differences in Table 3 were solely due to a pharmacist. That was the time that was overlapping with the Covid pandemic. How did that affect you folks?

Author Response

Comment 1: The fact that the post-intervention era the median Charlson Comorbidity Index was 5 and the proportion in ICU decreased compared to pre, can we say that the post-intervention cohort was less sick than pre-cohort.- Is this the reason why we see improvement in certain parameters in Table 3.

Response 1: Thank you for pointing this out, we mentioned that in the discussion section in page 8, line 265, The patient cohorts were not entirely comparable, highlighting differences in infection site, infection type, and patient location (p = 0.007, p < 0.001, and p = 0.004, respectively). 

Comment 2: Also there was no statistical difference in those discharged on antibiotics and readmission in the post cohort. How do you explain that?

Response 2: Thank you for pointing this out, I included in the discussion section, page 8, line 295;

Surprisingly, there was no statistically significant difference in the percentage of healthcare associated CDI, percentage of patients discharged on antibiotics or IRRR between the two periods. Several factors might have contributed to this finding. The sample size, while larger in the post-implementation group, may have been insufficient to detect subtle differences in readmission rates. Additionally, the shift in patient population towards more community-acquired infections and acute care admissions in the post-implementation group could also have independently influenced readmission rates. Finally, other complex factors beyond antibiotic use, like patient comorbidities and post-discharge care, likely play a significant role in readmissions, potentially overshadowing the impact of the AMS interventions alone. However, AMS pharmacists may contribute to IRRR by promoting appropriate use of antimicrobials, educating patients on medications adherence importance and follow-up care coordination.

Comment 3: Could the section on Cost Benefit Analysis be expanded?

Response: Thank you, I included more detailed results, Page 6, line 173;

Implementing weekend AMS pharmacist coverage led to a substantial reduction in overall costs. During the pre-implementation period, cost savings primarily resulted from antimicrobial de-escalation interventions, averaging $277 per patient. Post-implementation, however, saw a shift in savings mechanisms. De-escalation from broad-spectrum to narrow-spectrum antimicrobials yielded an average saving of $157 per patient, while discontinuation of unnecessary vancomycin contributed a significant $202 saving per patient. Overall, the average cost savings observed during the post-implementation period reached $28,493, a considerable increase compared to the $4,432 saved during the pre-implementation period.

and in the discussion section, page 9, line 311; We observed a considerable increase in cost savings from $4,432 in the pre-implementation period to $28,493 post-implementation, representing a 542.9% increase. This difference can be attributed to a shift in the types of interventions driving cost savings. Pre-implementation savings primarily stemmed from de-escalation, averaging $277 per patient. Post-implementation, de-escalation to narrower-spectrum antibiotics yielded $157 per patient, while discontinuation of unnecessary vancomycin emerged as a major driver, saving $202 per patient. It is important to acknowledge that these figures represent a conservative estimate, primarily reflecting direct cost reductions associated with drug acquisition. Quantifying the economic benefits of improved patient outcomes, such as reduced adverse events and shorter lengths of stay, poses a significant challenge. Further research exploring the broader economic landscape, including reduced complications, shorter hospital stays, and improved healthcare system efficiency, has the potential to demonstrate the true magnitude of cost savings associated with pharmacist driven AMS programs.

Comment 4: I find it difficult to prove that all the differences in Table 3 were solely due to a pharmacist. That was the time that was overlapping with the Covid pandemic. How did that affect you folks?

Response 4: Thank you, We excluded Covid patients and discussed other factors that contributed into our results in page 8, line 278; As a quaternary care hospital, we treat mostly complex cases with multiple comorbidities, several initiatives were implemented to reduce LOS between 2020 and 2021. These included building indication-specific order sets to guide physicians in ordering the most appropriate empiric antimicrobial therapy, implementing 72-hour antibiotic time-outs, building an automatic infectious disease consult upon prescribing certain restricted antimicrobials, referring patients to the outpatient parenteral antibiotic therapy (OPAT) at the infusion therapy center, discharging patients with home nursing services, transferring stable patients to long-term care nursing facilities, and implementing a daily case manager-led multidisciplinary team meeting to identify and resolve barriers to timely patient discharge. Despite these initiatives, the reported LOS reduction can be partially attributed to the pharmacist-led AMS service.

Also modified line 292; While a significant reduction in LOS was observed, this can be partially attributed to the AMS interventions, which, by optimizing antibiotic use, likely contributed to faster resolution of infections and facilitated more timely discharges. 

Kindly find the edited manuscript attached, all changes requested by the three reviewers are highlighted in yellow

Reviewer 3 Report

Comments and Suggestions for Authors

Thank you for the opportunity to review tjhis manuscript describing the benefits of adding weekend coverage by an ID trained clinical pharmacist to your institution's antimicrobial stewardship program. The study is well designed, with appropriate pre- and post-implementation criteria. I agree with your selection of primary and secondary outcomes and analytic methods. Your reesults are quite impressive, including cost savings. The discussion is well written, and appropriately compares your results to previous studies, and you address study limitations appropriately. 

I have only two questions:

1) Please clarify the size of the pre- and post-intervention populations. TABLE 1 shows respective pre and post sizes of 108 and 362. However, the first sentence under Results, and the second sentence of the paragraph discussing study limitations, indicate that the pre-intervdention population consisted of 362 subjects.

2) The post-intervention number of dose optimizations decreased significantly, which at first was counter-intuitive to me, since pharmacist coverage was extended by the additional weekend hours. Please address this further in your discussion. Is this the result of more prescribers consulting with the pharmacist prior to writing the initial order? If weekend coverage resulted in greater visibility and acceptance of clinical pharmacy consultations by weekend medical staff, this could be considered an additional advantage of the program.

Author Response

Comment 1: Please clarify the size of the pre- and post-intervention populations. TABLE 1 shows respective pre and post sizes of 108 and 362. However, the first sentence under Results, and the second sentence of the paragraph discussing study limitations, indicate that the pre-intervention population consisted of 362 subjects.

Response 1: Thank you for pointing this out, this was an error. Therefore, this was modified in the results section, page 3, line 113, to (The pre-implementation phase was composed of 108 subjects while the post implementation phase was composed of 362 subjects. Demographics and baseline characteristics were not significantly different between both groups (Table 1).)

Comment 2: The post-intervention number of dose optimizations decreased significantly, which at first was counter-intuitive to me, since pharmacist coverage was extended by the additional weekend hours. Please address this further in your discussion. Is this the result of more prescribers consulting with the pharmacist prior to writing the initial order? If weekend coverage resulted in greater visibility and acceptance of clinical pharmacy consultations by weekend medical staff, this could be considered an additional advantage of the program.

Response 2: Thank you for your comment, the discussion part was revised to clarify this point, page 6, line 217, to (While dose optimization interventions decreased (64% to 7.1%, p < 0.001) post-implementation, this is attributed to most initial dose adjustments being performed by clinical pharmacists during order verification, a process that precedes AMS pharmacist review, the AMS pharmacist may further adjust antimicrobial doses later based on changes in patients’ condition, these adjustments are less frequent than the initial optimizations performed at order verification. These findings collectively demonstrate a shift towards a more comprehensive AMS approach, optimizing not just the dose, but also the choice, duration, and appropriateness of antimicrobial therapy.)

Kindly find the edited manuscript attached, all changes requested by the three reviewers are highlighted in yellow

Round 2

Reviewer 2 Report

Comments and Suggestions for Authors

Thanks for addressing my concerns. The section on Cost benefit Analysis is extremely well detailed. Can you please indicate the method/s you used with some references preferably from UAE. 

I agree that the pre and post cohort aren't exactly comparable (as it is evident from Table 1). How do you think that affected your results?

I agree that definitely your pharmacist led initiative in addition to other multidisciplinary efforts had an impact on reducing infections. Can you add the multidisciplinary efforts as well. 

Author Response

Thank you for your insightful comments and suggestions on our manuscript. We appreciate your feedback and have revised the manuscript to address your concerns.

Concern 1: The section on Cost-benefit Analysis is extremely well-detailed. Can you please indicate the method/s you used with some references preferably from UAE?

Response: We acknowledge the lack of a detailed methodology for cost-benefit analysis in the initial manuscript. As this was a preliminary analysis focusing primarily on direct drug acquisition cost savings, we did not employ a specific established method. However, we recognize the need for a more robust approach to capture the full economic impact. Page 5, line 181

"We estimated direct drug cost savings by comparing the average daily cost of antibiotic therapy per patient before and after implementation, using our hospital's formulary pricing for 2020 (pre-implementation) and 2021 (post-implementation). This approach isolates cost differences attributable to pharmacist interventions, such as de-escalation and discontinuation of unnecessary therapy"

Unfortunately, literature on cost-benefit analysis methods specifically applied to AMS interventions in the UAE is limited, we have revised the discussion section to acknowledge this limitation and emphasize the need for future research utilizing comprehensive methodologies, Page 8, line 315.

"While a comprehensive cost-benefit analysis is beyond the scope of this retrospective study, and literature on cost-benefit analysis methods specifically applied to AMS interventions in the UAE is limited, our findings suggest a positive economic impact associated with the pharmacist-led weekend AMS program. "

Concern 2: I agree that the pre and post cohort aren't exactly comparable (as it is evident from Table 1). How do you think that affected your results?

Response: We acknowledge the difference in baseline characteristics between the pre and post-implementation cohorts. I added a paragraph in the discussion section elaborating on the potential influence of these differences on our findings, particularly on length of stay (LOS). Page 7, line 264.

"The observed reduction in length of stay (LOS) is likely influenced by a combination of factors. While the AMS intervention likely played a role in optimizing antibiotic use and accelerating infection resolution, it’s essential to acknowledge that potential confounding variables might have influenced the results. Notably, the pre- and post-implementation cohorts exhibited significant differences in patient characteristics (Table 1). This includes variations in infection site (p = 0.007), infection type (p < 0.001), and patient location (p = 0.004), which could independently influence LOS. For example, the pre-implementation group had a higher proportion of patients with bacteremia, often requiring longer treatment. Conversely, the post-implementation group had a higher prevalence of infections associated with shorter hospital stays, such as urinary tract, intra-abdominal, and skin and soft tissue infections. Furthermore, the shift towards community-acquired infections and increased admissions to acute care units in the post-implementation period may also have influenced LOS. While a significant reduction in LOS was observed, attributing the entire effect solely to the AMS intervention would be an overgeneralization. It is important to note that this reduction likely reflects a combination of factors, including optimized antibiotic use, improved discharge planning, and other multidisciplinary efforts."

Concern 3: I agree that definitely your pharmacist-led initiative in addition to other multidisciplinary efforts had an impact on reducing infections. Can you add the multidisciplinary efforts as well?

Response: We agree that our pharmacist-led initiative was one of several factors contributing to the observed outcomes. We have included a paragraph in the discussion section outlining the multidisciplinary efforts implemented during the study period to reduce LOS and optimize antimicrobial use. Page 7, line 281. 

"It is crucial to acknowledge that our pharmacist-led AMS initiative was one of several factors contributing to the observed outcomes. A multidisciplinary approach was employed during the study period to reduce length of stay and optimize antimicrobial use. This included initiatives such as the development of indication-specific order sets to guide physicians in prescribing the most appropriate empiric antimicrobial therapy, the implementation of 72-hour antibiotic timeouts to prompt reassessment of antibiotic necessity and facilitate timely discontinuation, automatic infectious disease consult referrals for restricted antibiotics to ensure expert review for high-risk cases, and outpatient parenteral antibiotic therapy (OPAT) referrals to allow eligible patients to transition to outpatient treatment. Additionally, discharge planning was refined with home nursing services to support patients requiring ongoing intravenous antibiotics at home, and stable patients were transferred to long-term care facilities, freeing up acute care beds and facilitating appropriate care transitions. Finally, daily multidisciplinary team meetings were implemented for discharge planning, fostering timely identification and resolution of discharge barriers. While a significant reduction in LOS was observed, this can be partially attributed to the multidisciplinary efforts implemented alongside the pharmacist-led weekend AMS service. These efforts synergistically contributed to optimizing antimicrobial use, improving patient outcomes, and facilitating more timely discharges."

Kindly find the attached modified manuscript with changes highlighted in green.
